# Differences in the optical properties of photochromic lenses between cold and warm temperatures

**Byeong-Yeon Moon, Sang-Yeob Kim, Dong-Sik Yu** *

Department of Optometry, Kangwon National University, Samcheok, Korea

* yds@kangwon.ac.kr

**Data Availability Statement:** All relevant data are within the manuscript and its Supporting Information files.

**Funding:** The author(s) received no specific funding for this work.

## Abstract

The aim of our study was to quantitatively evaluate the optical properties of photochromic lenses available on the market under cold and warm temperatures corresponding to the winter and summer seasons. The transmittance of 12 photochromic lenses from five manufacturers was measured using an UV/VIS spectrophotometer at cold (6 ± 2˚C) and at warm (21 ± 2˚C) temperatures. Transmittances were recorded from 380 to 780 nm and at the wavelength with maximum absorbance, which was calculated from the transmittance. The characteristics of the lenses were evaluated by examining changes in the optical properties at colorless and colored states and in the fading rate depending on temperature. The wavelength with maximum absorbance for photochromic lenses at the cold temperature showed a shorter shift than that at the warm temperature. The photochromic properties at the cold temperature were 11.5% lower for transmittance, 1.4 times higher for the change in optical density, and 1.2 times higher for the change in transmittance in the colored and colorless states, optical blocking % ratio, and change in luminous transmittance as compared to those at the warm temperature in the colored state. The fading rates based on the half-life time at the cold temperature were from 2.7 to 5.4 times lower than those at the warm temperature. The fading time until 80% transmittance was 6.4 times longer at the cold as compared to that at the warm temperature. There were significant differences in the optical properties of the photochromic lenses in terms of an absorbance at a shorter wavelength, a lower transmittance, a higher optical density, optical blocking % ratio, and luminous transmittance at the cold as compared to the warm temperature. Hence, it is necessary to provide consumers with information on photochromic optical properties, including the transmittance in colored and colorless states, and the fading rates at temperatures corresponding to the summer and winter seasons for each product.

## Introduction

Photochromic lenses are spectacle lenses involving a light-induced reversible change of color, i.e., they darken on exposure to ultraviolet (UV) rays in the presence of outdoor sunlight and

**Competing interests:** The authors have declared that no competing interests exist.

return to their clear state in the absence of activating light under indoor conditions [1, 2]. These lenses are used for reducing glare discomfort [3] and disability [4], improving photostress recovery [5], and protecting the eyes from UV radiation [6, 7]. The first commercial photochromic lenses were released by Corning Glass Works in 1964 [1], and various photochromic lenses have further been developed, with numerous manufacturers active in the present photochromic market, such as Carl Zeiss Meditec AG, Essilor International S.A., Hoya Corporation, Transitions Optical Limited, Rodenstock GmbH, Nikon Lenswear, and others [8]. Despite the ongoing advanced development, photochromic lenses have advantages and disadvantages compared to sunglasses [9]. The advantages include wearing convenience and continuous UV protection both indoors and outdoors, and the disadvantages pertain to the colored and colorless states of various degrees depending on the manufacturer, and unchangeable colors inside cars with UV blocking glass. These advantages and disadvantages are important factors influencing the consumer selection of photochromic lenses [3, 10].

Recently, manufacturers have made improvements in photochromic lens technology, such as the casting (or in-mass) process [11], to produce various lens designs of high index and improved photochromic performance for consumer satisfaction compared to other technologies, such as imbibing [12] and coating [13]. Today's photochromic lenses, however, are still being manufactured using the imbibing process, in which photochromic dyes are dispersed uniformly and deeply and the removed, or the coating process in which the dyes are coated evenly on the surface of the lenses, but the coated surfaces can be scratched. Many photochromic lenses currently on the market are being sold by eye care professionals based on information provided by the suppliers, including information on the refractive power, refractive index, center thickness, transmittance, and color [14]. As a result, it is challenging for the consumer, and even the eye care professional to understand the characteristics of photochromic lenses, including their transmittance at colored and colorless states and activating and fading rates. Moreover, because many manufacturers claim that their products are superior to those of the competitiors, comparisons of photochromic lenses among manufacturers are also challenging and it is difficult to locate the respective product features. Under these conditions, it is necessary to evaluate the characteristics of several photochromic lenses commonly available on the market [15–17].

A disadvantage of photochromic lenses is that they fade more slowly than they darken, within 20–30 s [4, 18]. A problem encountered by consumers is the long time required for the photochromic lens to completely fade when they move from outdoors (colored state) to indoors (colorless state). Therefore, the fading rate is an important factor in photochromic lens selection. In our previous studies [19–21], we investigated the optical characteristics and fading rates of photochromic lenses prepared by hard coatings and of marketed photochromic lenses, and suggested that manufacturers should provide consumers and agents with correct photochromic information regarding the fading rate. These fading rates in the above studies were evaluated at a temperature of $21 \pm 2°C$, similar to the $23 \pm 2°C$ of ISO 8980–3 [22]. Moreover, it is well known that photochromic lenses are darker at lower temperatures and that fading rates are longer when the lenses are colder [1, 23]. However, despite the growing use of photochromic lenses, studies on their characteristics, including the fading rate, in relation to temperature, which consumers or eye care professionals should be aware of, are lacking.

The present study aimed to evaluate the effects of temperature on the performance of the photochromic lenses supplied to the South Korean marketplace. Our interests were to determine the optical properties and fading rates at a cold temperature of $6 \pm 2°C$, similar to the mean temperature of $-6–7°C$ in January during the Korean winter, and at a warm temperature of $21 \pm 2°C$, similar to the mean temperatures of $23–27°C$ in August during the Korean summer [24].

## Materials and methods

### Materials

For a representative selection of commonly available plastic photochromic lenses in the South Korean market, we ordered photochromic lenses with brown and gray color (six of each) that changed from a relatively very light tint to a very dark tint under light conditions. Twelve photochromic lenses from five manufacturers were collected from Korean optical shops within 2 weeks. All lenses had a refractive index of 1.5–1.55 (middle index), were 70 mm in diameter, had plano with no refracting power (0.00 D), and had multicoated plastic photochromic lenses. Lens thickness was measured individually by a thickness gauge (ID-S1012; Mitutoyo, Kawasaki, Japan) and identified based on the specifications on the lens package or in an enclosed document. In addition, the manufacturing method of loading the photochromic materials was confirmed by reference to the manufacturer catalogue or directly by grinding the surface in the case of the coating method. The specifications of the photochromic lenses are shown in Table 1. Four lenses were manufactured by the imbibing, four by the casting, and four by the coating method.

### Measurement procedures

All lenses before the measurement of the spectral transmittance (transmittance) were stored in a sealed black box at room temperature. After the 12 photochromic lenses had underwent careful cleaning by cotton swab wetted with ethanol, their transmittance was measured using an UV/VIS spectrophotometer (X-ma 2000; Human Corporation, Seoul, Korea) with 190–900 nm wavelength, a deuterium and tungsten lamp as the light source, and a spectral bandwidth of 0.1–5.0 nm. The target temperature was set at either cold ($6 \pm 2$°C) or warm ($21 \pm 2$°C). The surrounding air in the laboratory room, chamber of the spectrophotometer, and the sealed box, used to create a dark environment, was controlled with an air cooler or air heater to maintain the target temperature.

To identify the colored and colorless states, the lenses at room temperature were activated by 10 pulses over 12 s in a photochromic lens tester (Quick; Nadokorea, Seoul, Korea) by the built-in UV pulse generation in a box (160 cm × 160 cm × 120 cm), and deactivated under

**Table 1. Specifications with plano of the photochromic lenses used in this study.**

| Specimen code[a] | Supplier | Refractive index[b] | Center thickness (mm)[c] | Color[b] | Manufacture process[b] |
|---|---|---|---|---|---|
| NKT gray | Nikon Lenswear Global | 1.5 | 1.98 | Gray | Imbibing |
| NKT brown | Nikon Lenswear Global | 1.5 | 1.92 | Brown | Imbibing |
| RDP gray | Rodenstock GmbH | 1.54 | 2.06 | Gray | Casting |
| RDP brown | Rodenstock GmbH | 1.54 | 2.07 | Brown | Casting |
| DMP gray | Daemyung Optical | 1.55 | 2.52 | Gray | Casting |
| DMP brown | Daemyung Optical | 1.55 | 2.53 | Brown | Casting |
| DMT gray | Daemyung Optical | 1.5 | 2.51 | Gray | Imbibing |
| DMT brown | Daemyung Optical | 1.5 | 2.50 | Brown | Imbibing |
| HYS gray | Hoya Corporation | 1.5 | 2.25 | Gray | Coating |
| HYS brown | Hoya Corporation | 1.5 | 2.30 | Brown | Coating |
| CZP gray | Carl Zeiss Meditec AG | 1.5 | 2.17 | Gray | Coating |
| CZP brown | Carl Zeiss Meditec AG | 1.5 | 2.16 | Brown | Coating |

[a]The first two characters signify the initials of company names and the next character indicates the initials of brand names; the last word refers to the colors.

[b]Specification identified on the lens package or supplier's catalogue and document.

[c]Mean value measured by a thickness gauge (ID-S1012; Mitutoyo, Kawasaki, Japan).

room illumination. The transmittance of the activated lenses was measured after they had been stored in a sealed black box for at least 12 h.

For evaluating the optical properties of the lenses, when the target temperature was reached under dim illumination, each lens was moved to the spectrophotometer set at 1-nm intervals and a 500 nm/min scanning speed. The transmittance of the colorless state was measured first. Second, to measure the transmittance of the colored state, the lens was placed in the photochromic lens tester and was activated by 10 pulses in 12 s. After the lens had quickly been moved to the spectrophotometer, the transmittances were recorded at the intervals of 0, 120, 240, and 360 s. If absorbance (A) was needed, it was calculated by A = 2 –log(T) as an equation of the relationship between A and transmittance (T, %) [2]. In addition, after storing for at least 12 h at room temperature, the transmittance was again measured at intervals of 0, 30, 60, 90, and 120 s at the wavelength with the maximum absorbance obtained from the previous measured transmittance.

## Optical properties

The optical properties of photochromic lenses were evaluated by $\lambda_{max1}$ as the wavelength with maximum absorbance in the colored state and the maximum difference in absorbance between the colored and colorless states when scanning at warm or cold temperature, by the transmittance in the colorless ($T_\infty$) and colored ($T_0$) states at $\lambda_{max1}$, by the $\triangle OD$ as the change in optical density expressed as $\log_{10}(T_\infty/T_0)$, by the $\triangle T_{max1}$ as the difference in transmittance between the colorless and colored states at $\lambda_{max1}$, by the $\triangle T_{mean}$ as the difference in the mean value of transmittance measured in the visible region, by the $BR_{max1}$ as the optical blocking % ratio of $\triangle T\%$ to the colorless state ($T_\infty$) at $\lambda_{max1}$, by the $BR_{mean}$ as the optical blocking % ratio of $\triangle T\%$ to the colorless state based on the mean value measured in the visible region, by the luminous transmittance of the colorless ($LT_\infty$) and the colored state ($LT_0$), and by the $\triangle LT$ as the difference in the luminous transmittance between the colorless and colored states. The luminous transmittance was calculated by the ratio of the luminous flux transmitted by the lens to the incident luminous flux [22].

## Fading rate

The switchable mechanism between the colored and colorless states is a reversible reaction [25–27]. In particular, the fading process is the first-order reaction accompanying a closed ring within photochromic materials [26, 28]. Hence, the fading rate can be evaluated based on the half-life time derived from the following equations (Eqs 1 and 2).

$$-\text{In} \frac{(A_t - A_\infty)}{(A_0 - A_\infty)} = kt \tag{1}$$

$$t_{1/2} = \frac{\text{In} 2}{k} \tag{2}$$

Here, $A_t$ and $A_0$ denote the absorbance at time t and time zero in the activation of the colored state, respectively. $A_\infty$ is the absorbance after the lens remained in the dark for at least 12 h, k is the rate constant in the fading process, and $t_{1/2}$ represents the half-life time. The absorbance derived from the above data of transmittance differed with the criteria of wavelength. Hence, we evaluated the fading rate based on the half-life time in three ways. The first was the half-life time $t_{1(1/2)}$ determined by $\lambda_{max1}$ as the wavelength with maximum absorbance in the colored state and the maximum difference in absorbance between the colored and colorless states when scanning at the warm or cold temperature. The second was the half-life time $t_{2(1/2)}$

determined by $\lambda_{max2}$ as the wavelength with the maximum difference in the absorbance between the colored and colorless states when scanning at the warm temperature. The third was the half-life time $t_{3(1/2)}$ determined by the mean value of the difference in absorbance between the colored and colorless states at the warm or cold temperature at 380–780 nm scanning.

In addition, the fading rate can be evaluated by the $T_{80\%}$ as the fading time until 80% transmittance, corresponding to very lightly tinted sun-glare filters, at $\lambda_{max1}$ is reached [20].

The fading rates of $t_{1(1/2)}$, $t_{2(1/2)}$, and $t_{3(1/2)}$ for photochromic lenses based on the half-life time determined at $\lambda_{max1}$, $\lambda_{max2}$, and the mean of 380–780 nm, respectively, were evaluated between cold and warm temperatures.

## Statistical analysis

All data were collected (S1 File) and were statistically analyzed using MedCalc (Version 12.7.7.0; MedCalc Software, Mariakerke, Belgium). The half-life time related to the fading rate was determined using Excel spreadsheets (S2 File). The Kolmogorov–Smirnov test was first performed to test for variable normality, and the Wilcoxon test was used for paired-sample comparisons, the Kruskal–Wallis test for comparisons among three or more groups, and Spearman's rank correlation coefficient ($\rho$) to test for associations between variables. A p-value $\leq 0.05$ was considered statistically significant.

## Results

### Optical properties of photochromic lenses

The optical properties of photochromic lenses at warm and cold temperatures are shown in Table 2. $\lambda_{max1}$ with maximum absorbance ranged from 571 nm to 592 nm in 11 of the 12 (six lenses at each temperature) gray photochromic lenses and 443 nm to 484 nm in 10 of the 12 brown photochromic lenses at warm and cold temperatures. As shown in Fig 1A–1D, $\lambda_{max1}$ of the brown photochromic lenses appeared mostly at a short wavelength (Fig 1B and 1D), and the absorbance bands of these lenses were found lower at long wavelengths and higher at short wavelengths than they were in gray photochromic lenses (Fig 1A and 1B). The maximum absorbance at the cold temperature shifted to an on average 7 nm shorter wavelength in the range of 560 nm to 585 nm and an on average 2 nm shorter wavelength in the range of 455 nm to 465 nm than it did at the warm temperature in gray and brown photochromic lenses.

The total mean transmittance of the colored state ($T_0$) at $\lambda_{max1}$ was 11.5% darker, i.e., 23.1% at cold temperatures compared to 34.6% at warm temperatures, in the range from 6% for the HYS gray lens to 28.9% for the DMP brown lens for the difference in transmittance between the two temperatures. However, Spearman's rank correlation coefficients of the transmittance (92.2 ± 3.4%) and thickness (2.25 ± 0.22 mm) were not significantly different (n = 24, $\rho$ = -0.217, p = 0.472). $\Delta OD$ was used to evaluate the difference in concentration between the colorless and colored states. The mean $\Delta OD$ was 0.639 (range: 0.321–0.966), 1.4 times higher at the cold than at the warm temperature 0.458 (range: 0.239–0.687). The difference in transmittance ($\Delta T_{max1}$) between the colorless and colored states at $\lambda_{max1}$, on average, was 68.4% (range: 47.5–83.4%) at the cold and 58.3% (range: 36.5–72.6%) at the warm temperature. The difference in the mean value of transmittance ($\Delta T_{mean}$) measured in the visible region, on average, was 42.0% (range: 30.0–51.0%) at the cold and 34.1% (range: 23.0–45.9%) at the warm temperature [20]. The $\Delta T$ ($\Delta T_{max1}$ and $\Delta T_{mean}$) at the cold temperature was 1.2 times higher than that at the warm temperature, being 10.1% higher at $\lambda_{max1}$ and 7.9% higher in the visible region. The BR was used to evaluate how well the photochromic lenses performed as anti-glare sunglasses. The $BR_{max1}$ at $\lambda_{max1}$, on average, was 74.6% (range: 52.3–89.2%) at the cold and 62.8%

**Table 2. Optical properties of photochromic lenses.**

| Specimen code | Temperature[a] | $\lambda_{max1}$ (nm) | $T_\infty{}^b$ (%) | $T_0{}^b$ (%) | $\Delta OD^c$ | $\Delta T_{max1}{}^d$ ($\Delta T_{mean}$)[e] (%) | $BR_{max1}{}^f$ ($BR_{mean}$)[g] (%) | $LT_\infty{}^h$ (%) | $LT_0{}^h$ (%) | $\Delta LT^i$ (%) ($LT_\infty - LT_0$) |
|---|---|---|---|---|---|---|---|---|---|---|
| NKT gray | Warm | 580 | 93.5 | 29.4 | 0.502 | 64.1 (35.9) | 68.6 (41.3) | 93.0 | 37.5 | 55.6 |
|  | Cold | 577 | 92.5 | 17.1 | 0.733 | 75.4 (45.8) | 81.5 (52.7) | 92.2 | 23.3 | 68.9 |
| NKT brown | Warm | 577 | 95.1 | 39.2 | 0.385 | 55.9 (32.1) | 58.8 (36.5) | 94.3 | 44.6 | 49.7 |
|  | Cold | 462 | 93.3 | 24.0 | 0.590 | 69.3 (43.0) | 74.3 (48.4) | 95.0 | 29.8 | 65.2 |
| RDP gray | Warm | 573 | 92.9 | 53.1 | 0.243 | 39.8 (23.0) | 42.8 (27.2) | 91.5 | 55.9 | 35.5 |
|  | Cold | 478 | 91.1 | 43.5 | 0.321 | 47.6 (32.1) | 52.3 (36.8) | 93.9 | 46.6 | 47.4 |
| RDP brown | Warm | 484 | 86.2 | 49.7 | 0.239 | 36.5 (23.2) | 42.3 (27.2) | 92.8 | 56.8 | 36.0 |
|  | Cold | 480 | 81.8 | 34.3 | 0.377 | 47.5 (30.0) | 58.1 (35.4) | 92.1 | 47.0 | 45.1 |
| DMP gray | Warm | 589 | 96.4 | 44.9 | 0.332 | 51.5 (28.4) | 53.4 (32.2) | 95.5 | 52.1 | 43.4 |
|  | Cold | 585 | 96.9 | 35.4 | 0.437 | 61.5 (38.0) | 63.5 (42.3) | 96.6 | 40.9 | 55.7 |
| DMP brown | Warm | 456 | 96.5 | 53.4 | 0.257 | 43.1 (24.0) | 44.7 (27.2) | 95.2 | 59.5 | 35.7 |
|  | Cold | 457 | 88.2 | 24.5 | 0.556 | 63.7 (35.4) | 72.2 (40.5) | 95.1 | 42.7 | 52.5 |
| DMT gray | Warm | 585 | 93.1 | 21.5 | 0.637 | 71.6 (44.6) | 76.9 (51.7) | 92.8 | 27.7 | 65.2 |
|  | Cold | 582 | 92.5 | 16.0 | 0.762 | 76.5 (49.9) | 82.7 (57.7) | 92.6 | 20.9 | 71.7 |
| DMT brown | Warm | 443 | 93.8 | 24.2 | 0.588 | 69.6 (40.3) | 74.2 (45.7) | 94.3 | 31.5 | 62.9 |
|  | Cold | 444 | 94.7 | 14.1 | 0.827 | 80.6 (46.2) | 85.1 (51.7) | 95.1 | 24.8 | 70.3 |
| HYS gray | Warm | 592 | 91.4 | 18.8 | 0.687 | 72.6 (45.9) | 79.4 (53.8) | 91.9 | 26.2 | 65.7 |
|  | Cold | 587 | 88.4 | 12.8 | 0.839 | 75.6 (51.0) | 85.5 (60.6) | 89.5 | 18.0 | 71.6 |
| HYS brown | Warm | 460 | 89.7 | 21.8 | 0.614 | 67.9 (40.6) | 75.7 (46.9) | 91.6 | 30.0 | 61.6 |
|  | Cold | 460 | 93.5 | 10.1 | 0.966 | 83.4 (50.5) | 89.2 (56.4) | 94.4 | 20.5 | 73.9 |
| CZP gray | Warm | 580 | 93.4 | 23.0 | 0.609 | 70.4 (39.2) | 75.4 (45.5) | 93.6 | 31.7 | 61.9 |
|  | Cold | 577 | 92.8 | 16.2 | 0.758 | 76.6 (44.9) | 82.5 (52.3) | 93.1 | 23.6 | 69.5 |
| CZP brown | Warm | 573 | 93.0 | 36.4 | 0.407 | 56.6 (32.1) | 60.9 (37.2) | 93.2 | 42.4 | 50.8 |
|  | Cold | 571 | 91.9 | 28.9 | 0.502 | 63.0 (36.8) | 68.6 (43.0) | 92.0 | 34.0 | 58.0 |
| All |  | Mean | 92.2 | 28.8 | 0.549 | 63.3 (38.0) | 68.7 (43.8) | 93.4 | 36.2 | 57.2 |
|  |  | SD | 3.4 | 13.1 | 0.205 | 13.3 (8.6) | 14.2 (9.9) | 1.6 | 12.5 | 12.1 |
|  | Warm | Mean | 92.9 | 34.6 | 0.458 | 58.3 (34.1) | 62.8 (39.4) | 93.3 | 41.3 | 52.0 |
|  |  | SD | 2.8 | 13.2 | 0.168 | 13.1 (8.2) | 14.2 (9.6) | 1.3 | 12.3 | 11.9 |
|  | Cold | Mean | 91.5 | 23.1 | 0.639 | 68.4 (42.0) | 74.6 (48.2) | 93.5 | 31.0 | 62.5 |
|  |  | SD | 3.9 | 10.5 | 0.204 | 12.0 (7.3) | 11.9 (8.4) | 1.9 | 10.8 | 10.2 |

SD: standard deviation; $\lambda_{max1}$: wavelength with maximum absorbance at the colored state and the maximum difference in absorbance between the colored and colorless states when scanning at warm or cold temperature.

[a]Warm temperature: 21 ± 2°C, cold temperature: 6 ± 2°C.

[b]Transmittance in colorless ($T_\infty$) and colored state ($T_0$) at $\lambda_{max1}$.

[c]$\Delta OD$: change in optical density is $\log_{10}(T_\infty/T_0)$.

[d]$\Delta T_{max1}$: the difference in transmittance between the colorless and colored states at $\lambda_{max1}$.

[e]$\Delta T_{mean}$: the difference in the mean value of transmittance measured in the visible region, and values at warm temperature are data derived from our previous study [20].

[f]$BR_{max1}$: optical blocking % ratio of $\Delta T$% to colorless state ($T_\infty$) at $\lambda_{max1}$.

[g]$BR_{mean}$: optical blocking % ratio of $\Delta T$% to colorless state based on the mean value measured in the visible region, and values at warm temperature are data derived from our previous study [20].

[h]Luminous transmittance of the colorless ($LT_\infty$) and the colored state ($LT_0$), respectively.

[i]$\Delta LT$: the difference in luminous transmittance between the colorless and colored states.

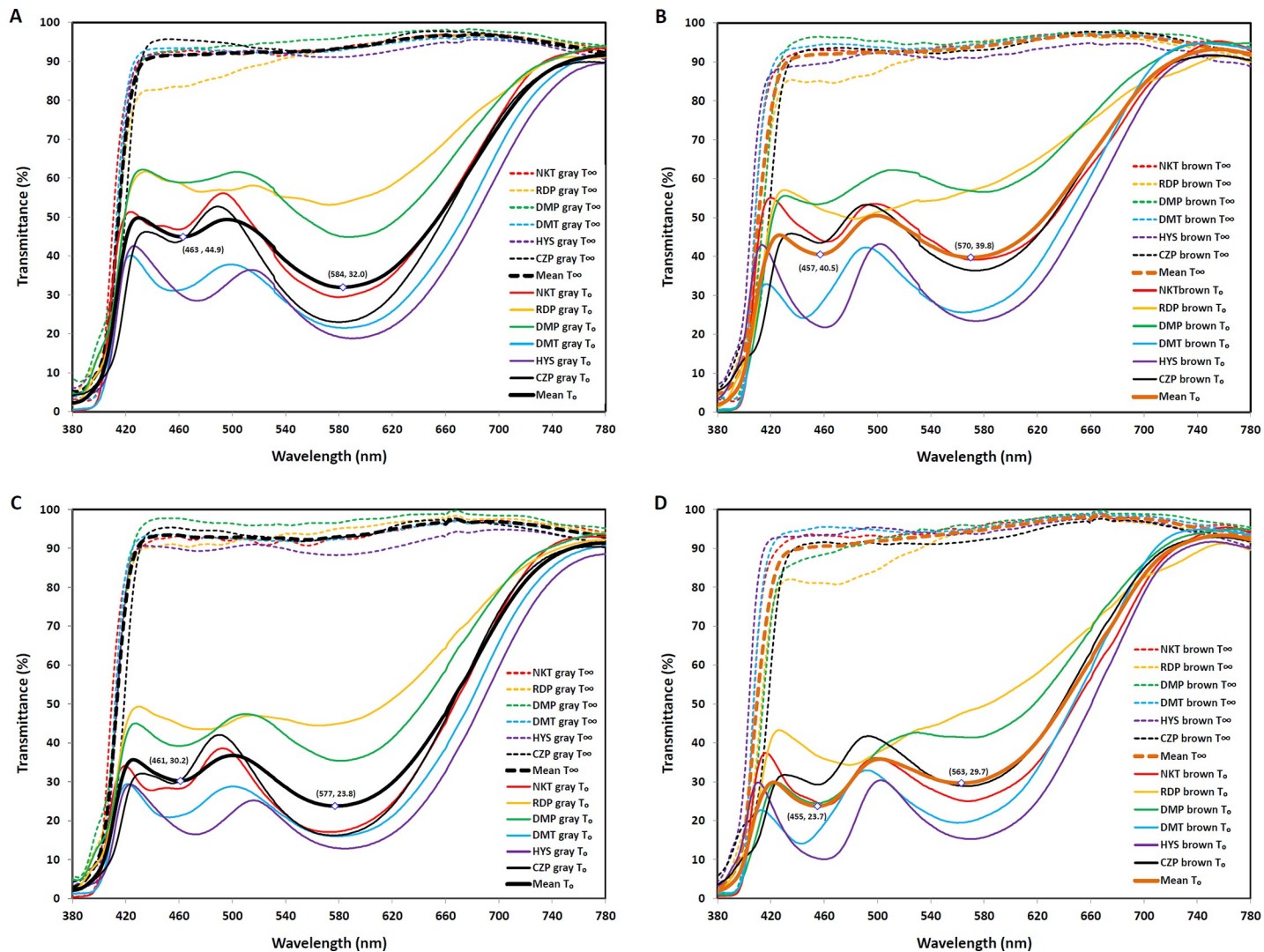

**Fig 1. Transmittance for photochromic lenses.** (A) Transmittance of gray photochromic lenses at the colored ($T_0$) and colorless state ($T_\infty$) at the warm temperature. (B) Transmittance of brown photochromic lenses at the colored ($T_0$) and colorless state ($T_\infty$) at the warm temperature. (C) Transmittance of gray photochromic lenses at the colored ($T_0$) and colorless state ($T_\infty$) at the cold temperature. (D) Transmittance of brown photochromic lenses at the colored ($T_0$) and colorless state ($T_\infty$) at the cold temperature.

(range: 42.3–79.4%) at the warm temperature. The $BR_{mean}$ evaluated in the visible region, on average, was 48.2% (range: 35.4–60.6%) at the cold temperature and 39.4% (range: 27.2–53.8%) at the warm temperature [20]. The BR ($BR_{max1}$ and $BR_{mean}$) at the cold temperature was 1.2 times higher than that at the warm temperature, being 11.8% higher at $\lambda_{max1}$ and 8.8% higher in the visible region. The difference in luminous transmittance between the colorless and colored states ($\Delta LT$) at the cold temperature was 62.5% (range: 45.1–73.9%), 1.2 times higher than the 52.0% (range: 35.5–65.7%) at the warm temperature.

## Correlation between optical properties of photochromic lenses

Spearman's rank correlation coefficients were analyzed to evaluate the relationship between transmittance ($T_\infty$ and $T_0$) and $\Delta OD$, $\Delta T$, and BR, showing the performance of photochromic lenses at the colorless and colored states. Spearman's rank correlation coefficients between $T_0$

**Table 3. Correlation coefficients between transmittance, optical blocking % ratio, and luminous transmittance.**

|  | $T_\infty$: $LT_\infty$ | $T_0$: $LT_0$ | $\triangle T_{max1}$: $\triangle LT$ | $\triangle T_{mean}$: $\triangle LT$ |
|---|---|---|---|---|
| All (n = 24) | 0.686 (0.001) | 0.969 (< 0.001) | 0.976 (< 0.001) | 0.988 (< 0.001) |
| Warm (n = 12) | 0.877 (0.004) | 0.986 (0.001) | 0.972 (0.001) | 0.970 (0.001) |
| Cold (n = 12) | 0.596 (0.048) | 0.895 (0.003) | 0.923 (0.002) | 0.965 (0.001) |
| Gray (n = 12) | 0.627 (0.038) | 0.993 (0.001) | 0.965 (0.001) | 0.965 (0.001) |
| Brown (n = 12) | 0.600 (0.047) | 0.944 (0.002) | 0.979 (0.001) | 0.984 (0.001) |
|  | $\triangle LT$: $BR_{max1}$ | $\triangle LT$: $BR_{mean}$ | $\triangle T_{max1}$: $BR_{max1}$ | $\triangle T_{mean}$: $BR_{mean}$ |
| All (n = 24) | 0.977 (< 0.001) | 0.981 (< 0.001) | 0.984 (< 0.001) | 0.990 (< 0.001) |
| Warm (n = 12) | 0.951 (0.002) | 0.972 (0.001) | 0.979 (0.001) | 0.991 (0.001) |
| Cold (n = 12) | 0.951 (0.002) | 0.944 (0.002) | 0.944 (0.002) | 0.965 (0.001) |
| Gray (n = 12) | 0.979 (0.001) | 0.965 (0.001) | 0.972 (0.001) | 1.000 (0.001) |
| Brown (n = 12) | 0.965 (0.001) | 0.991 (0.001) | 0.972 (0.001) | 0.996 (0.001) |

(): Significance level p in Spearman's rank correlation coefficient.

and ΔOD, ΔT, and BR were significant (p < 0.001 for all), but they were not significant between $T_\infty$ and ΔOD, ΔT, and BR (p = 0.831 for ΔOD, p = 0.793 for $\Delta T_{max1}$, p = 0.864 for $\Delta T_{mean}$, p = 0.831 for $BR_{max1}$, and p = 0.618 for $BR_{mean}$). The results showed that $T_0$ was an important factor and better able to reveal the optical properties of the photochromic lenses than $T_\infty$.

Other correlations were analyzed to examine the applicability of transmittance instead of luminous transmittance (LT) weighted by the photopic spectral sensitivity of the human eye at each wavelength. These Spearman's rank correlation coefficients are presented in Table 3. The correlations between T and LT were significant (ρ = 0.596–0.877, p = 0.001–0.048 for $T_\infty$ versus $LT_\infty$; ρ = 0.895–0.993, p ≤ 0.03 for $T_0$ versus $LT_0$), and the correlations between ΔT and ΔLT were also significant (ρ = 0.923–0.979, p ≤ 0.002 for $\triangle T_{max1}$ versus $\triangle LT$, 0.965–0.988, p ≤ 0.001 for $\triangle T_{mean}$ versus $\triangle LT$). ΔLT also tended to have a closer relationship with $BR_{mean}$ than with $BR_{max1}$, and BR, exhibiting a high correlation with $\triangle LT$ was more strongly correlated with $\triangle T_{mean}$, based on the mean values in the visible region, than to $\triangle T_{max1}$ based on $\lambda_{max1}$.

## Fading rate of photochromic lenses

The fading rate based on the half-life time was calculated using Eq 2 expressed as the rate constant (k) determined from the plotting of time versus $-\ln(A_t - A_\infty)/(A_0 - A_\infty)$, as, for example, shown in Fig 2A–2D. The figures show a linearity between time, and the logarithm of absorbance showed the following order: cold temperature at $\lambda_{max2}$, warm temperature at $\lambda_{max2}$, cold temperature at $\lambda_{max1}$, and warm temperature at $\lambda_{max1}$. Although the evaluation of linearity is limited by the measurement of only a few points, a lower linearity in the photochromic lenses, including NKT gray, existed at the warm temperature at $\lambda_{max1}$. This linearity is, in part, due to the scanning range (from 780 to 380 nm) over a long period (0–360 sec). Scanning may cause a difference between the time-intervals at each $\lambda_{max1}$, and the measurement over a long relative to a short time may be also influence the linearity. The fading rate measured at $\lambda_{max1}$, $\lambda_{max2}$, and the mean of 380–780 nm at cold and warm temperatures is shown in Table 4, and the fading rate ($t_{3(1/2)}$) at the warm temperature was calculated in a previous study [20].

The Kolmogorov–Smirnov test showed that the fading rate according to the half-life time was not normally distributed (p = 0.017). In comparing the fading rate based on the half-life time, the fading rate was 2.7 times longer for $t_{1(1/2)}$ (Wilcoxon test for paired samples, p = 0.001), 5.4 times longer for $t_{2(1/2)}$ (Wilcoxon test for paired samples, p < 0.001), and 3.3

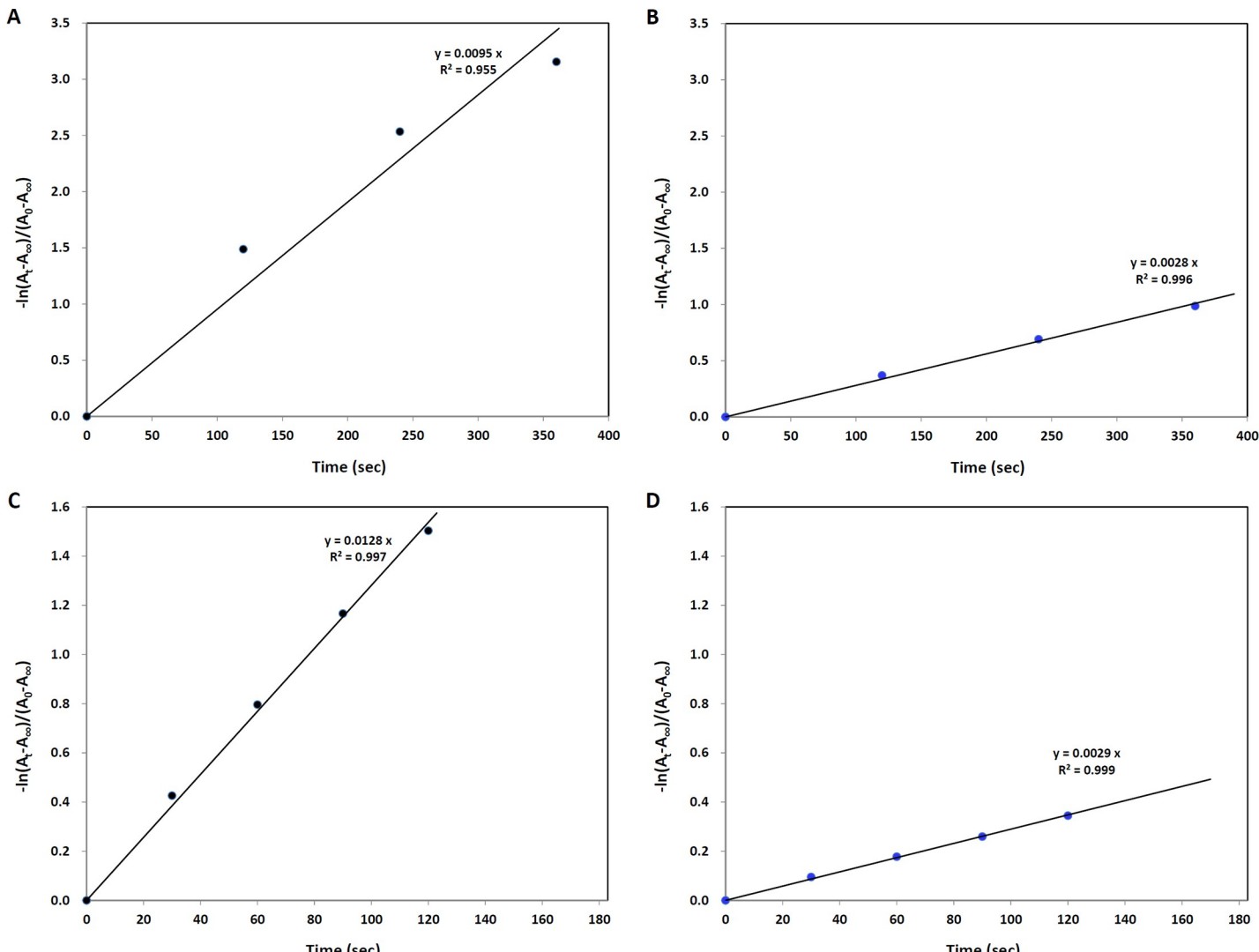

**Fig 2. Examples for plotting time versus $-\ln(A_t - A_\infty)/(A_0 - A_\infty)$ in determining the fading rate constant (k).** (A) A plot for an NKT gray photochromic lens at $\lambda_{max1}$ at the warm temperature. (B) A plot for an NKT gray photochromic lens at $\lambda_{max1}$ at the cold temperature. (C) A plot for an NKT gray photochromic lens at $\lambda_{max2}$ at the warm temperature. (D) A plot for an NKT gray photochromic lens at $\lambda_{max2}$ at the cold temperature.

times longer for $t_{3(1/2)}$ (Wilcoxon test for paired samples, $p < 0.001$) at the cold than at the warm temperature. The fading rates of photochromic products varied from 63 s to 198 s for $t_{1(1/2)}$, from 38 s to 72 s for $t_{2(1/2)}$, and from 44 s to 147 s for $t_{3(1/2)}$ at the warm temperature and from 186 s to 335 s for $t_{1(1/2)}$, from 222 s to 447 s for $t_{2(1/2)}$, and from 210 s to 408 s for $t_{3(1/2)}$ at the cold temperature. However, the Kruskal–Wallis test showed that there were no significant differences among the three fading rates ($t_{1(1/2)}$, $t_{2(1/2)}$, and $t_{3(1/2)}$) at both temperatures ($p = 0.255$). The fading rate based on $T_{80\%}$ also was 6.4 times longer at the cold than at the warm temperature (Wilcoxon test for paired samples, $p < 0.001$). The Wilcoxon test between the three fading rates and $T_{80\%}$ showed that there were significant differences at both temperatures ($p < 0.001$). Spearman's rank correlation coefficient between the three fading rates and $T_{80\%}$ showed 0.772 ($p < 0.001$) for $t_{3(1/2)}$ and $T_{80\%}$, 0.781 ($p < 0.001$) for $t_{1(1/2)}$ and $T_{80\%}$, and 0.946 ($p < 0.001$) for $t_{2(1/2)}$ and $T_{80\%}$. Based on these correlation analyses, $t_{2(1/2)}$ was best at showing the colored state.

**Table 4. Fading rates based on half-life time measured at $\lambda_{max1}$, $\lambda_{max2}$, and the mean of 380–780 nm.**

| Specimen code | Temperature | $\lambda_{max1}$ | $k_1$, $\times 10^{-3}$ | $t_{1(1/2)}$ (sec) | $\lambda_{max2}$ | $k_2$, $\times 10^{-3}$ | $t_{2(1/2)}$ (sec) | $k_3$, $\times 10^{-3}$ | $t_{3(1/2)}$ (s)[a] | $T_{80\%}$ (s) |
|---|---|---|---|---|---|---|---|---|---|---|
| NKT gray | Warm | 580 | 9.5 | 73 | 590 | 12.8 | 54 | 9.5 | 73 | 222 |
|  | Cold | 577 | 2.8 | 247 | 590 | 2.9 | 239 | 2.4 | 289 | 1051 |
| NKT brown | Warm | 577 | 8.7 | 79 | 580 | 11.5 | 60 | 8.2 | 85 | 224 |
|  | Cold | 462 | 2.4 | 287 | 580 | 2.1 | 329 | 2.2 | 315 | 1439 |
| RDP gray | Warm | 573 | 5.3 | 132 | 596 | 11.7 | 59 | 5.7 | 122 | 207 |
|  | Cold | 478 | 2.3 | 306 | 596 | 3.2 | 214 | 2.3 | 301 | 1108 |
| RDP brown | Warm | 484 | 3.8 | 181 | 572 | 10.9 | 63 | 4.7 | 147 | 232 |
|  | Cold | 480 | 2.6 | 265 | 572 | 3.0 | 235 | 2.5 | 277 | 1142 |
| DMP gray | Warm | 589 | 7.4 | 93 | 591 | 12.9 | 54 | 6.6 | 105 | 181 |
|  | Cold | 585 | 2.9 | 240 | 591 | 2.8 | 247 | 2.7 | 257 | 802 |
| DMP brown | Warm | 456 | 3.5 | 198 | 584 | 13.5 | 51 | 5.8 | 120 | 146 |
|  | Cold | 457 | 3.7 | 186 | 584 | 3.3 | 222 | 3.2 | 217 | 652 |
| DMT gray | Warm | 585 | 8.3 | 83 | 596 | 12.5 | 55 | 8.2 | 85 | 253 |
|  | Cold | 582 | 2.5 | 283 | 596 | 1.9 | 363 | 2.1 | 330 | 1501 |
| DMT brown | Warm | 443 | 8.0 | 87 | 570 | 9.6 | 72 | 7.9 | 88 | 243 |
|  | Cold | 444 | 2.3 | 297 | 570 | 1.8 | 383 | 2.0 | 347 | 1317 |
| HYS gray | Warm | 592 | 9.1 | 76 | 595 | 11.3 | 61 | 9.0 | 77 | 265 |
|  | Cold | 587 | 2.1 | 335 | 595 | 1.6 | 442 | 1.7 | 408 | 2196 |
| HYS brown | Warm | 460 | 11.1 | 63 | 462 | 11.0 | 63 | 15.6 | 44 | 275 |
|  | Cold | 460 | 2.2 | 306 | 462 | 1.6 | 447 | 1.9 | 365 | 2949 |
| CZP gray | Warm | 580 | 8.6 | 81 | 582 | 18.1 | 38 | 7.9 | 88 | 124 |
|  | Cold | 577 | 3.6 | 193 | 582 | 2.8 | 247 | 1.9 | 365 | 944 |
| CZP brown | Warm | 573 | 8.5 | 82 | 575 | 18.1 | 38 | 7.1 | 98 | 138 |
|  | Cold | 571 | 3.6 | 193 | 575 | 2.8 | 247 | 3.3 | 210 | 979 |

| | | $t_{1(1/2)}$ | $t_{2(1/2)}$ | $t_{3(1/2)}$ | $T_{80\%}$ |
|---|---|---|---|---|---|
| All, mean ± SD (sec) | | 182 ± 93 | 178 ± 139 | 201 ± 118 | 775 ± 731 |
|  | Warm | 102 ± 44 | 56 ± 10 | 94 ± 27 | 209 ± 51 |
|  | Cold | 262 ± 50 | 301 ± 87 | 307 ± 61 | 1340 ± 646 |

$\lambda_{max1}$: wavelength with maximum absorbance at the colored state and the maximum difference in absorbance between the colored and colorless states when scanning at warm or cold temperature; $\lambda_{max2}$: wavelength with the maximum difference in absorbance between the colored and colorless states when scanning based on warm temperature; $k_1$ and $k_2$: fading rate constant at $\lambda_{max1}$ and $\lambda_{max2}$, respectively; $t_{1(1/2)}$ and $t_{2(1/2)}$: half-life time at $\lambda_{max1}$ and $\lambda_{max2}$, respectively; $t_{3(1/2)}$: half-life time at mean transmittance in the visible region; $T_{80\%}$: fading time until 80% transmittance at $\lambda_{max1}$ is reached

[a]Values at the warm temperature are derived from our previous study [20].

## Time-related changes in absorbance in determining the rate constant (k)

The fading rates based on half-life time are determined by the rate constant (k), which reflects time-related changes in absorbance. The coefficient of determination ($R^2$) related to k is presented in Table 5. All $R^2$ values except for the RDP brown and DMP brown lenses were higher than 0.900. These high $R^2$ values clearly explain the time-related changes in absorbance (or transmittance). In the correlation between the $R^2$ values, the $\rho$ between $R1^2$ and $R3^2$ was 0.818 (p < 0.001) higher than the 0.681 (p = 0.001) for $R2^2$ and $R3^2$, and 0.717 (p = 0.001) for $R1^2$ and $R2^2$. In the relative comparison of $R^2$, there was a higher $R^2$ at the cold than at the warm temperature and at $R2^2$ than at $R1^2$ and $R3^2$.

**Table 5. Comparison of coefficients of determination as predictors of time-related changes in absorbance in determining the rate constants.**

| Specimen code | Temperature | R1$^2$ | R2$^2$ | R3$^2$ |
|---|---|---|---|---|
| | | $R^2$ at $\lambda_{max1}$ (In scanning) | $R^2$ at $\lambda_{max2}$ (In fixing) | $R^2$ at 380–780 nm[a] (In scanning) |
| NKT gray | Warm | 0.955 | 0.997 | 0.972 |
| | Cold | 0.996 | 0.999 | 0.999 |
| NKT brown | Warm | 0.943 | 0.997 | 0.946 |
| | Cold | 0.994 | 1.000 | 0.998 |
| RDP gray | Warm | 0.924 | 0.952 | 0.930 |
| | Cold | 0.973 | 0.985 | 0.972 |
| RDP brown | Warm | 0.860 | 0.943 | 0.887 |
| | Cold | 0.976 | 0.986 | 0.971 |
| DMP gray | Warm | 0.958 | 0.979 | 0.941 |
| | Cold | 0.982 | 0.997 | 0.982 |
| DMP brown | Warm | 0.755 | 0.983 | 0.867 |
| | Cold | 0.974 | 0.998 | 0.986 |
| DMT gray | Warm | 0.984 | 0.995 | 0.993 |
| | Cold | 1.000 | 0.999 | 1.000 |
| DMT brown | Warm | 0.982 | 1.000 | 0.993 |
| | Cold | 0.999 | 0.999 | 1.000 |
| HYS gray | Warm | 0.996 | 1.000 | 1.000 |
| | Cold | 1.000 | 1.000 | 1.000 |
| HYS brown | Warm | 0.999 | 1.000 | 0.900 |
| | Cold | 1.000 | 0.995 | 0.999 |
| CZP gray | Warm | 0.986 | 0.999 | 0.986 |
| | Cold | 0.999 | 1.000 | 1.000 |
| CZP brown | Warm | 0.974 | 0.998 | 0.952 |
| | Cold | 0.999 | 1.000 | 1.000 |
| All, mean ± SD | | 0.967 ± 0.055 | 0.992 ± 0.015 | 0.970 ± 0.039 |
| | Warm | 0.943 ± 0.071 | 0.987 ± 0.020 | 0.947 ± 0.044 |
| | Cold | 0.991 ± 0.011 | 0.996 ± 0.005 | 0.992 ± 0.011 |

SD: standard deviation

$\lambda_{max1}$: wavelength with maximum absorbance at the colored state and the maximum difference in absorbance between the colored and colorless states when scanning at the warm or cold temperature.

$\lambda_{max2}$: wavelength with the maximum difference in absorbance between the colored and colorless states when scanning based on the warm temperature.

$R^2$: coefficient of determination as the prediction of time-related changes in absorbance.

[a]Values at the warm temperature are derived from our previous study [20].

## Discussion

Photochromic lenses act as sunglasses in that they change their tint depending on the weather or the presence or absence of UV radiation, but they are used throughout the year, not only during the summer season. In the current study, the characteristics of photochromic lenses supplied to the South Korean marketplace were evaluated at warm (21 ± 2˚C) and cold (6 ± 2˚C) temperatures, closely approximating temperatures during the Korean summer and winter, as a factor affecting photochromism. The changes in the performance of photochromic lenses included a shorter wavelength shift of maximum absorbance, a lower transmittance in the colored state, and a slower fading rate at cold than at warm temperatures. These changes were compared and evaluated quantitatively.

## Optical properties and their relationship

As shown in Fig 1, $\lambda_{max1}$ with maximum absorbance (minimum transmittance) was shifted to a shorter wavelength at the cold in contrast to the warm temperature. This result is similar to the previously reported finding that the maximum absorbance of photochromic spiropyran appeared at a slightly shorter wavelength with decreasing temperature [29]. In another study, however, the absorption band of photochromic naphthopyran showed a very slight shift to a shorter wavelength as the temperature increased [30]. In photochromic lenses, materials such as oxazines, pyrans, and fulgides are added to plastic lens material with polarity, such as PMMA, CR39, polycarbonate, and polyurethane [1, 31]. In this case, the shift of the maximum absorbance of photochromic materials is affected by the structure of the materials as well as the polarity and flexibility of matrices such as polymethyl methacrylate (PMMA), and the absorption of spiropyran under the influence of polarity may cause a shift to a shorter wavelength [32]. Several studies have shown that the wavelength shift of the maximum absorbance for photochromic lenses depends on temperature, photochromic materials, matrix polarity, and other environmental conditions. In this study, photochromic lenses with polarity showed a shorter wavelength shift with decreasing temperature.

Photochromic lenses are temperature and thickness dependent. We found that temperature affects the transmittance of photochromic lenses. As shown in Table 2, the transmittance at the colored state ($T_0$) was on average 11.5% darker at the cold temperature than at the warm temperature. However, as a general rule, thicker photochromic lenses may darken to a somewhat greater degree compared to thinner ones [33, 34]. However, the correlation of transmittance with thickness was not statistically significant. This finding signifies that there was no difference in thickness among the photochromic lenses.

The performance of photochromic lenses as sunglasses and general spectacles is evaluated as the photochromic response, in which the ratio of the luminous transmittance of a photochromic specimen in its faded state and, after 15 min irradiation, in its darkened state shall be at least 1.25 [22]. However, the performance of photochromic lenses could not be fully reflected as the photochromic response is the least requirement. For this reason, various optical properties of photochromic lenses were evaluated in this study, and we found that temperature affected several of these. $\Delta OD$ and $\Delta T$ in our results were 1.4 and 1.2 times higher at the cold than at the warm temperature, respectively. The BR and $\Delta LT$ were also 1.2 times higher at the cold than at the warm temperature. The optical properties of photochromic lenses such as $\Delta OD$, $\Delta T$, and BR, are factors consisting of $T_\infty$ and $T_0$. Therefore, these factors will be affected by $T_\infty$ and $T_0$. $\Delta LT$ is the difference between $LT_\infty$ and $LT_0$. If LT ($LT_\infty$ and $LT_0$) is related to $T_\infty$ and $T_0$, then $\Delta LT$ will be also affected by $T_\infty$ and $T_0$. To determine the main factors revealing the optical properties of photochromic lenses, Spearman's rank correlation coefficients between the transmittance ($T_\infty$ and $T_0$) and $\Delta OD$, $\Delta T$, BR, and LT were calculated. In the correlation analysis, $T_0$ was found to be a more important factor than $T_\infty$ in revealing the optical properties of photochromic lenses. In addition, from the correlations between transmittance ($\triangle T$) and LT and BR, and between BR and LT, both $\Delta LT$ and BR were more strongly correlated to $\triangle T_{mean}$ than $\triangle T_{max1}$, and the relationship between BR with $\Delta LT$ was stronger for $BR_{mean}$ than in $BR_{max1}$. $\triangle T_{mean}$ and $BR_{mean}$ based on the mean values in the visible region were more important parameters than $\triangle T_{max1}$ and $BR_{max1}$ based on $\lambda_{max1}$ in evaluating the optical properties of the photochromic lenses. Therefore, the main factors in evaluating the optical properties of the photochromic lenses were $T_0$, $\triangle T_{mean}$, and $BR_{mean}$.

It would be reasonable to evaluate the effects of photochromic lenses on human visual performance by luminous transmittance [34], which differs among colored lenses, instead of measuring transmittance by spectrophotometry. The fading rate, however, cannot be directly

measured by luminous transmittance, considering the eye's sensitivity to each wavelength instead of transmittance. If transmittance is closely related to luminous transmittance, it will be possible to use transmittance to evaluate photochromic lenses. From the correlation analysis shown in Table 3, the high correlations of $T_0$ and $LT_0$, $\triangle T$ and $\triangle LT$, and $\triangle LT$ and BR signified that luminous transmittance can be replaced by transmittance in evaluating the performance of photochromic lenses. Therefore, as ophthalmic lenses are characterized by their transmittance [34], the optical properties of photochromic lenses could also be evaluated by transmittance instead of luminous transmittance.

## Comparison of fading rates determined based on half-life time

In our study, the fading rates were evaluated based on the half-life time. Our results showed that the fading rates in the solid matrix for a difference of approximately 15˚C were 2.7 to 6.4 times longer at the cold than at the warm temperature, as shown in Fig 2 and Table 4. The fading rates decreased at the cold temperature. Megla [35] also reported that the fading rate depends on temperature. In another study [36], the fading rate of naphthoxazine in a common organic solvent was reported to increase three times for every 10˚C increase in temperature. Although our experiment was not performed below 0˚C, the fading rate at −6˚C can be approximately 2–3 times longer than that at $6 \pm 2$˚C when considering the warm versus cold temperature ratios for $k_1$ and $k_2$ in Table 4, as also shown in the study of Chu [36]. Large differences in the half-life time measured at $\lambda_{max1}$ and $\lambda_{max2}$ were evident for RDP brown, DMP brown, HYS gray, and HYS brown. The $t_{2(1/2)}$ at the warm temperature was shorter than $t_{1(1/2)}$ in RDP brown and DMP brown, having a higher transmittance (low absorbance) (Fig 1B). These differences may be due to the differences between $\lambda_{max1}$ and $\lambda_{max2}$ in the scanning range (from 780 to 380 nm). However, $t_{1(1/2)}$ at the cold temperature was shorter than $t_{2(1/2)}$ in HYS gray and HYS brown. These lenses also showed a low transmittance (high absorbance) at the cold temperature (Figs 1A and 2B). The differences were more noticeable at the cold than at the warm temperature, which may be due to the properties of the photochromic materials [26, 35] and the matrix polarity [30, 37] in the lenses. However, in the present study, this is difficult to explain because there was no information on the composition of HYS gray and brown, such as related to photochromic dyes and the matrix. The temperature dependence was lower for RDP gray, RDP brown, and DMP brown for both $k_1$ and $k_2$, and was higher for HYS gray and HYS brown for both $k_1$ and $k_2$. It was high for NKT brown for $k_1$ and DMT gray for $k_2$. The relationship between the ratio of the warm to the cold temperature for $k_1$ and $k_2$ was significant for Spearman's rank correlation ($\rho = 0.705$, $p = 0.019$). There were no statistically significant differences among the three fading rates ($t_{1(1/2)}$, $t_{2(1/2)}$, $t_{3(1/2)}$) determined by different methods in this study. However, the differences between fading rates of photochromic products show various distributions. Comparing our results with those of other studies [30, 35, 37], the fading rate of photochromic materials appears to depend on the photochromic specimen, temperature, and photochromic dye–matrix or solvent interaction.

The fading rates based on the half-life time are determined by the rate constant (k) as first-order reaction mechanisms in photochemical processes [26, 28]. The coefficient of determination ($R^2$) is an important quantity that evaluates how well a rate constant explains a fading rate in the first-order reaction of time and absorbance. As shown in Table 5, the $R^2$ related to the rate constant (k) in determining the fading rate of the photochromic lenses was higher at $\lambda_{max2}$ than at $\lambda_{max1}$ and at 380–780 nm scanning, and higher at the cold than at the warm temperature. Therefore, the fading rates were better determined and explained by $\lambda_{max2}$ and the cold temperature.

Spearman's rank correlation coefficient between the three half-life times related to the fading rate and $T_{80\%}$ were higher for $t_{2(1/2)}$ than for $t_{1(1/2)}$ and $t_{3(1/2)}$. From these results, the half-

**Table 6. Characteristics of each process in determining the half-life time.**

| Criterion | Process | Exclusion of variance factors | | |
|---|---|---|---|---|
| | | Scanning time | Temperature | Luminous transmittance |
| $\lambda_{max1}$ in scanning | In scanning, 1) Wavelength at maximum absorbance in the colored state and 2) Maximum difference in absorbance between the colored and colorless states at warm and cold temperatures, respectively. | No | Yes | No |
| $\lambda_{max2}$ on fixing | On fixing, 1) Wavelength at maximum difference in absorbance between the colored and colorless states at only the warm temperature. 2) Wavelength at the warm temperature is applied to the cold temperature. | Yes | No | No |
| Mean of 380 to 780 nm | In 380–780 nm scanning, the mean value of the difference in absorbance between the colored and colorless states at warm and cold temperatures. | No | Yes | Yes |

life time of $t_{2(1/2)}$ was best at showing the colored state. The fading rates also depend on the wavelength criteria in the process of determining the half-life time except for $T_{80\%}$. In this study, the delay time in transferring the activated lens to the spectrophotometer was not considered, but a relative comparison of the characteristics of each fading rate is considered possible. In addition, the fading rates were limited to their relative comparison based on transmittance without considering the manufacturing process. Consequently, further studies are needed to establish a method for determining the fading rates based on the luminous transmittance of human eyes and, furthermore, to assess whether the differences in fading rates between cold and warm temperatures affect photochromic lens wearer satisfaction, such as vision-related quality of life [7, 38]. Even so, characteristics of each process in determining the fading rate in the current study can be summarized as shown in Table 6. In the analysis of the characteristics of each process in determining the half-life time related to the fading rate, a good process is to minimize the variance of absorbance over time, to indicate the difference in absorbance over temperature, and to maximize the effect of luminous transmittance. The process to achieve this involves determining $\lambda_{max}$ at a given temperature in the transmittance region (a near wavelength of 550 nm), which well reflects the luminous transmittance, and to determine the half-life time at $\lambda_{max}$ in a fixed state without scanning from 780 nm to 380 nm.

From our findings, however, the optical properties of photochromic lenses in the colored and colorless states varied by manufacturer, color, and temperature. Information regarding these characteristics should be clearly known in the market to increase wearer satisfaction [10].

In summary, this study evaluated changes in the optical properties of photochromic lenses available on the market between cold and warm temperatures closely resembling summer and winter weather as a factor affecting photochromism. Changes in the performance of photochromic lenses between colored and colorless states were clearly indicated and included a shorter wavelength shift with maximum absorbance, a lower transmittance in the colored state, a higher OD, a higher optical blocking % ratio, and a higher luminous transmittance at the cold than at the warm temperature. Moreover, the fading rate at the cold temperature was 2.5–5.4 times longer than at the warm temperature. It is currently not known how these differences between the two temperatures affect photochromic lens wearer perception and satisfaction. However, the optical properties of photochromic lenses available on the market varied by temperature and product. Therefore, as the temperature according to the season affects the performance of the photochromic lens, it is necessary to provide consumers with accurate information regarding the colored state and fading rate as photochromic characteristics are affected by the summer and winter season for each product.

## Supporting information

**S1 File. Raw data for photochromic lenses.**
(XLSX)

**S2 File. Spreadsheets for determination of half-life time of Fig 2.**
(XLSX)

## Author Contributions

**Conceptualization:** Byeong-Yeon Moon, Dong-Sik Yu.

**Data curation:** Byeong-Yeon Moon, Dong-Sik Yu.

**Formal analysis:** Byeong-Yeon Moon, Sang-Yeob Kim, Dong-Sik Yu.

**Investigation:** Byeong-Yeon Moon.

**Methodology:** Byeong-Yeon Moon, Sang-Yeob Kim, Dong-Sik Yu.

**Project administration:** Dong-Sik Yu.

**Resources:** Byeong-Yeon Moon.

**Supervision:** Dong-Sik Yu.

**Validation:** Sang-Yeob Kim, Dong-Sik Yu.

**Visualization:** Sang-Yeob Kim.

**Writing – original draft:** Byeong-Yeon Moon, Sang-Yeob Kim, Dong-Sik Yu.

**Writing – review & editing:** Dong-Sik Yu.

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
