## [Decision Letter · Decision Letter 0]

7 Feb 2020

PONE-D-19-32995

Differences in the optical properties of photochromic lenses between cold and warm temperature

PLOS ONE

Dear Mr. Yu,

Thank you for submitting your manuscript to PLOS ONE. After careful consideration, we feel that it has merit but does not fully meet PLOS ONE’s publication criteria as it currently stands. Therefore, we invite you to submit a revised version of the manuscript that addresses the points raised during the review process.

The manuscript has been evaluated by three reviewers, and their comments are available below.

The reviewers have raised a number of concerns that need attention. They request additional information on methodological aspects of the study and have made several points be clarified in the text.

Could you please revise the manuscript to carefully address the concerns raised?

We would appreciate receiving your revised manuscript by Mar 23 2020 11:59PM. To enhance the reproducibility of your results, we recommend that if applicable you deposit your laboratory protocols in protocols.io, where a protocol can be assigned its own identifier (DOI) such that it can be cited independently in the future. For instructions see: http://journals.plos.org/plosone/s/submission-guidelines#loc-laboratory-protocols

We look forward to receiving your revised manuscript.

Kind regards,

George Vousden

Senior Editor

PLOS ONE

Journal Requirements:

Reviewers' comments:

Reviewer's Responses to Questions

**Comments to the Author**

1. Is the manuscript technically sound, and do the data support the conclusions?

Reviewer #1: Yes

Reviewer #2: Partly

Reviewer #3: Yes

2. Has the statistical analysis been performed appropriately and rigorously? 

Reviewer #1: Yes

Reviewer #2: I Don't Know

Reviewer #3: Yes

3. Have the authors made all data underlying the findings in their manuscript fully available?

Reviewer #1: Yes

Reviewer #2: Yes

Reviewer #3: Yes

4. Is the manuscript presented in an intelligible fashion and written in standard English?

Reviewer #1: Yes

Reviewer #2: No

Reviewer #3: Yes

5. Review Comments to the Author

Reviewer #1: It is well known that photochromic lenses darken more at medium and low temperatures (temperatures above zero). The conclusions of the research carried out by the authors confirm this phenomenon (see table 2. Optical properties of photochromic lenses./ Mean values of luminous transmittance for all tested samples (LT0)at warm temperature is 41% and in cold temperature is 31,0 %). The authors tested lenses conditioned at positive temperatures (6 and 12 oC). The choice of these temperatures is dictated by the climatic conditions in which protective glasses with mounted photochromic lenses are used. It would be interesting to carry out these tests for lenses conditioned at temperatures well below freezing. The practical aspect of the research is very important. I fully agree with the authors that information on changes in the selected optical properties of photochromic lenses (e.g. optical density, colored state and fading rate) should also be provided to users of glasses.

Suggestion to introduce small additions to the text:

The Authors write: “Other correlations were analyzed to examine the applicability of transmittance instead of luminous transmittance (LT) weighted by the photopic spectral sensitivity of the human eye at each wavelength”. To determine the value of luminous transmittance, the illuminant distribution is also taken into account. I would suggest writing the right formula in the Optical properties section or referring to the distribution of the illuminant in the text.

Reviewer #2: This paper reports an analysis of the photochromic behavior of commercial photochromic lenses with a focus on the effect of temperature on the performance of these systems. The authors use a big collection of gray and brown lenses from different manufacturers.

The analysis of the results is quite detailed but the discussion is rather disorganized and difficult to follow. In particular, it would be very useful to add comparative graphics to follow the discussion which, for 16 samples, has just too many numbers. Also, the results and discussion parts should be divided into subsections and each section should include an objective conclusion.

The quality of the few available graphics is very low and they are impossible to read in the pdf file.

The fading kinetics of the photochromic lenses was assumed to follow a mono-exponential behavior which is a big simplification since two colored species are formed. From fig 2A one can see the effects of this simplification: the data does not follow a straight line. Fig 2b,c,d show smaller deviations because the authors collected only 4 points, which is rather low. This aspect should be commented.

The experiments were conducted at 21 degrees to simulate the Korean summer and at 6 degrees to simulate the Korean winter (-6-7 degree). Although I understand the difficulties in performing this experiments at -6 degrees, the authors should comment on the expected effect from going from 6 to -6 degrees.

Some of the lenses show a big difference in the lifetime measured at lmax1 or lmax 2. This may lead to a color change during decoloration. For instance, DMP brown has a lifetime of 198 s at 456 nm and 51 s at 584 nm. This is quite different and must be commented. The same for RDP brown.

The conclusions of this work are rather vague. That the fading kinetics are temperature depend everyone knows. For instance, the authors don’t identify which lenses show the lowest temperature dependency, which one maintains its color shade, or not. After a so long statistical study, the conclusions are a deception. Since the goal was to analyze the marketed lenses the study should be more objective especially the conclusions.

The conclusion could include a table or a paragraph to answer all the topics written in the last paragraph of the abstract:

Abstract: “There were significant differences in the optical properties of photochromic lenses in terms of showing absorbance at a shorter wavelength, lower transmittance, higher optical density, higher optical blocking % ratio, and higher luminous transmittance in cold temperature than in warm temperature. Hence, it is necessary to provide consumers with information on photochromic optical properties including …”

Reviewer #3: The authors present a study on the kinetics of commercially available photochromic spectacle lenses. The study seems to be well conducted and the results have been clearly presented. The underlying data has been made available.

Specific comments:

1. The statement in the abstract does not seem to be correct "... properties at cold than at warm temperatures were 11.5% lower for transmittance at the colored state, 1.4 times higher for optical density". According to the data this should be "1.4 times higher for the change in optical density (from faded to colored state).

The corresponding text in the manuscript main text should be revised accordingly

2. page 5, line 107: "All were middle index" does not sound very scientific. As according to the given refractive indices of the lenses, most of the lenses are most likely made of CR-39, the standard material. I'd suggest the following: "All lenses had a refractive index of 1.5-1.54 (standard material)".

3. page 6, line 128: Please add some more data on the spectrometer, what kind of light source was used?

4. page 9, line 193: is this correct that there is three times t1(1/2) ?? Does this make any sense?

5. The fact that photochromism shows different behaviour related to the ambient temperature has been shown previously. The authors may want to include the following literature to provide a better unterstanding on the thermal effects in photochromic materials:

R. J. Araujo, "Kinetics of Bleaching of Photochromic Glass," Appl. Opt. 7, 781-786 (1968)

D. C. Look and W. L. Johnson, "Transmittance of photochromic glass at environmental extremes," Appl. Opt. 18, 595-597 (1979)

6. The authors should discuss, that the measurements are affected by the fact that the activation has been performed outside the spectrometer and the lens was already deactivating/fading during spectrometer measurement. This may affect the results considering that the scanning speed of the spectrometer (500 nm/min) is relatively long. Given that only a VIS-spectrum was measured (bandwidth ~ 380 nm between 380 nm and 760 nm), the measurement would take about 46 seconds which is nearly half of t(1/2) at least for the measurement at the warm state

7. In general it would be extremel interesting to also have the change in the UV transmittance/absorbance of the lenses (< 380 nm). Are these data available?

Good luck with your manuscript!

6. PLOS authors have the option to publish the peer review history of their article (what does this mean?). If published, this will include your full peer review and any attached files.

Reviewer #1: No

Reviewer #2: No

Reviewer #3: No

---

## [Author Response · Author response to Decision Letter 0]

29 Feb 2020

These responses are in a attach file related Response to Editor’ comments and Response to Reviewers' comments:

We have revised our manuscript accordingly and provided point-by-point responses to the reviewers’ comments.

All the authors express their appreciation for your kind consideration of our manuscript. 

I look forward to hearing from you.

Sincerely,

---

## [Decision Letter · Decision Letter 1]

19 May 2020

Differences in the optical properties of photochromic lenses between cold and warm temperatures

PONE-D-19-32995R1

Dear Dr. Yu,

We are pleased to inform you that your manuscript has been judged scientifically suitable for publication and will be formally accepted for publication once it complies with all outstanding technical requirements.

With kind regards,

Timo Eppig

Guest Editor

PLOS ONE

Additional Editor Comments (optional):

I participated as a reviewer for the initial evaluation of this manuscript.

Reviewers' comments:

Reviewer's Responses to Questions

**Comments to the Author**

1. If the authors have adequately addressed your comments raised in a previous round of review and you feel that this manuscript is now acceptable for publication, you may indicate that here to bypass the “Comments to the Author” section, enter your conflict of interest statement in the “Confidential to Editor” section, and submit your "Accept" recommendation.

Reviewer #1: (No Response)

Reviewer #2: All comments have been addressed

2. Is the manuscript technically sound, and do the data support the conclusions?

Reviewer #1: (No Response)

Reviewer #2: Yes

3. Has the statistical analysis been performed appropriately and rigorously? 

Reviewer #1: (No Response)

Reviewer #2: Yes

4. Have the authors made all data underlying the findings in their manuscript fully available?

Reviewer #1: (No Response)

Reviewer #2: Yes

5. Is the manuscript presented in an intelligible fashion and written in standard English?

Reviewer #1: (No Response)

Reviewer #2: Yes

6. Review Comments to the Author

Reviewer #1: (No Response)

Reviewer #2: (No Response)

7. PLOS authors have the option to publish the peer review history of their article (what does this mean?). If published, this will include your full peer review and any attached files.

Reviewer #1: No

Reviewer #2: No

---

## [Editor Report · Acceptance letter]

21 May 2020

PONE-D-19-32995R1 

Differences in the optical properties of photochromic lenses between cold and warm temperatures 

Dear Dr. Yu:

I am pleased to inform you that your manuscript has been deemed suitable for publication in PLOS ONE. Congratulations! Your manuscript is now with our production department. 

With kind regards,

on behalf of

Dr. Timo Eppig 

Guest Editor

PLOS ONE